# Ousting the Cypriot *Ethnarch*: President Makarios' Struggle against the Greek Junta, Cypriot Bishops, and Terrorism

**Şevki Kıralp** [1,*] **and Ahmet Güneyli** [2]

1   Faculty of Economics and Administrative Sciences, Cyprus International University, Nicosia 99258, Cyprus
2   Education Faculty, European University of Lefke, Lefke 99010, Cyprus; aguneyli@eul.edu.tr
*   Correspondence: skiralp@ciu.edu.tr

**Abstract:** This study examined the politics and political involvements of the Orthodox Church of Cyprus in the early postcolonial era, with a special focus on the ecclesiastical coup that aimed for the ouster of Archbishop Makarios III, who was also the President of the Republic of Cyprus from 1960 to 1977. The findings indicate that the Greek junta, Greek Cypriot terrorists, and the three bishops of the Orthodox Church of Cyprus joined forces to oust Makarios by forcing him to resign his presidency. These actors were displeased with Makarios because he tolerated Cypriot communism, refused to follow Athens' manipulations in Cypriot politics, and promoted Cyprus' independence by abandoning the pro-Enosis (unification of Cyprus with Greece) political line. The Greek junta tried to dictate policies to Makarios and asked him to resign as he refused to obey. Greek Cypriot terrorists engaged in violence to destabilize the island and oust Makarios. The three bishops summoned the Holy Synod and defrocked the Archbishop as he refused to resign his presidency. Importantly, this research came across with strong indicators that the Greek junta tried to utilize religion in trying to oust the Cypriot *ethnarch* as the three bishops, immediately after the junta's failure to oust Makarios in 1972, asked him to resign his presidency. While his rivals failed to oust Makarios, at least until 1974, he called for an international synod and defrocked the three bishops. He managed to retain both posts until the end of his life.

**Keywords:** Cyprus; Makarios; orthodoxy; Holy Synod

## 1. Introduction

The understanding of nationalism and identity in Cypriot communities, the international and domestic aspects of the Cyprus conflict, and the modern and early–modern history of the island have attracted a significant amount of scholarly attention. Any study focused on the aforementioned topics have hardly neglected the historical and political role of the Greek Cypriot *ethnarchy* in Cypriot history and politics. Furthermore, Archbishop Makarios' understanding of politics, his confrontations with Athens and Grivas, and his *ethnarchic* authority have also been fairly covered by the literature. Nevertheless, the three Cypriot bishops' attempts to oust Makarios, namely, the ecclesiastical coup that commenced in February 1972 and ceased when the international Supreme Synod that defrocked these bishops in July 1973, have, to some extent, remained under-studied. It is well-known in the literature that Makarios' political power was mostly sourced from the traditional authorities granted to Cypriot *ethnarchs*. Nonetheless, his main rivals in the Greek and Greek Cypriot political scenes (namely, the Greek junta and Grivas) also utilized religion in their attempts to oust Makarios as they leagued with the three bishops, and this remains considerably neglected. This study intended to contribute to the literature on this very point. As this specific State–Church conflict was, at the same time, an intra-church conflict, as Makarios was also the Head of the State, this research will likely be useful in making wider sense of State–religion relations in Cyprus. This paper examined this church–*ethnarchy* conflict and concludes that Grivas and the Greek junta made an unsuccessful

attempt to oust Makarios by utilizing the opposition of the three bishops against their Archbishop.

For data gathering, this study utilized British national archives (TNA), American national archives (NARA: online access), Greek Cypriot and Turkish newspapers, the Cypriot Press and Information Office's (PIO) press releases archive (online access), and statemen's interviews and memoirs. The findings indicate that Makarios was challenged by a league composed of the Greek junta, Greek Cypriot terrorists, and three bishops of the Orthodox Church of Cyprus. The research conducted indicates that the aforementioned actors joined forces against Makarios, as he did not tolerate Greek interferences in Cypriot affairs and abandoned the pro-Enosis political line. The Greek junta was considerably displeased with Makarios' refusal to follow Athens' manipulations in Cypriot politics. Greek Cypriot terrorist organizations were composed of extreme nationalists dissatisfied with Makarios' abandonment of the pro-Enosis line. In 1972 and 1973, the three aforementioned bishops tried to make Makarios resign his presidency and defrocked him when he refused. As they were part of this anti-Makarios league, it is difficult to claim that the bishops had purely religious and non-political purposes. This league tried to oust Makarios and based their arguments on both religious and political factors. The Greek junta overthrew Makarios by using military force in 1974. Nevertheless, the attempts to force Makarios to resign his presidency had commenced long before the military coup of 1974. Makarios was a leader who simultaneously held the presidency and the archbishopric. He countered the bishops' decision by calling for an international synod and managed to remain in both offices until the end of his life (3 August 1977).

## 2. The Orthodox Church of Cyprus in the Island's Political History

In 395, the Roman Empire accepted Christianity as the State's official religion and placed the church under two heads: The West (Rome) and the East (Constantinople). The fall of the Western flank of the Roman Empire in the fifth century made this split even sharper (Streeter 2012, pp. 158–60). Centuries before the Great Schism of 1054, there had been disagreements between Latin- and Greek-speaking churches, particularly on the source of the Holy Spirit. At odds with the Latin-speaking Western churches, the Greek-speaking Eastern churches believed that the Holy Spirit was sourced exclusively by the Father and not by the Son. Alongside such different interpretations, the political rivalries and cultural differences between the West and the East created the basis of the Catholic–Orthodox split. In 1054, the Pope and the Patriarch of Constantinople mutually excommunicated one another. Consequently, the Greek-speaking churches of the East (Byzantium) embraced Orthodoxy (Tickle 2012). The Church of Cyprus was founded in 45 A.D. by Paul the Apostle and Saint Barnabas (McBirnie 2013). In 431, it gained autonomy from the see in Antioch (Kidd 2010, p. 331) and it is one of the oldest autocephalous Eastern Orthodox churches. It is institutionally led by the Holy Synod, comprising the Archbishop; the bishops of Kitium, Kyrenia, Limassol, Morphou, and Paphos; the suffragan bishops of Salamis, Trimithous, and Arsinoe; and the bishop of Kykkos (Mirbagheri 2009, p. 32). The Archbishop of the Orthodox Church of Cyprus is elected by the Holy Synod. The church comprises four episcopal sees: The archbishopric of Nicosia and the bishoprics of Paphos, Kitium, and Kyrenia. It is ruled according to the written constitution prepared in 1909 (Roudometof 2013, pp. 103–12). The Orthodox Church of Cyprus has 11 active monasteries, and some of them date back to the Byzantium era. It regards itself of equal rank with the patriarchates of Constantinople, Jerusalem, Antioch, and Alexandria. It recognizes the Ecumenical Patriarchate's primacy in theological matters (McGuckin 2010). The fairly decentralized structure of the Orthodox Church of Cyprus enables Cypriot clergymen to promote various attitudes toward Church–State relations. Three groups prevail among Cypriot clergymen, namely, the *ethnarchikoi*, *organosiakoi*, and *paterikoi*. The *ethnarchikoi* believe that the church should retain its economic power and interfere in political issues concerning the Cyprus issue. They regard themselves as followers of Makarios. Meanwhile, the *organosiakoi* oppose openness to the Catholic and Protestant churches, and the *paterikoi*

promote the theology of the "Greek Fathers of the Church". All three groups cooperate with one another within the Church, yet they retain their differences (Sarris 2010, pp. 197–200).

The Ottoman Empire conquered Constantinople in 1453 and destroyed the Byzantium. Nevertheless, Ottomans benefited from the Catholic–Orthodox split with regard to inhibiting the Christian reunion. To this end, they granted the Orthodox Church large authorities and refrained from exerting religious pressure on its Orthodox population (Ortaylı and Armağan 2004). On the contrary, prior to the fall of Constantinople in the mid-15th century, the Grand Duke of Byzantium would rather "see a Turkish turban in the midst of the City than the Latin mitre" (Baş 2018). When Cyprus was conquered by Ottomans in 1571, the Orthodox people of the island were pleased, as they became free of their Catholic rulers. The Orthodox Church of Cyprus was generally loyal to Istanbul. For instance, in 1804, the Church and the Ottoman Governor cooperated to suppress a riot that had been launched by Turkish elites and supported by Greek and Turkish rebellions (Kızılyürek 2002).

In the 18th Century Ottoman Cyprus, the archbishopric was devoted the title "*Ethnarch*" and accepted as the political leader and representative of Greek Cypriots by Istanbul (Katsourides 2017, p. 100). As the other non-Muslim communities in Ottoman territories, Greek Cypriots were subject to their own rules rather than the rules of Muslims. The *ethnarchs* were obliged to remain loyal to Istanbul and to assure their communities' loyalty as well. Accordingly, the Cypriot Archbishop had significant authorities, including tax collection (Şahin 1980). On some occasions, Greek Cypriot *ethnarchs* even managed to convince Istanbul to make significant changes in the administrative system of Cyprus. In 1821, Greece launched its war of independence against Ottomans. The Cypriot Archbishop Kyprianos was accused of instigating Greek Cypriots to rebel, and he was sentenced to death along, with the bishops of Paphos, Kyrenia, and Kitium (Göktürk 2015). One perspective is that there was no convincing evidence that Kyprianos really conspired to instigate anti-Ottoman activities in Cyprus and that the Ottoman Governor might have wanted to destroy the Archbishop, as he perceived the latter to be a political rival (Michael 2005, pp. 215–40). Since the Greek struggle for independence, unification of Cyprus with Greece (Enosis) had been a national inspiration for Greek Cypriots. Nonetheless, the massive mobilization of Cypriot Hellenism for Enosis happened in the British era. Essentially, leading the struggle for Enosis became an ethno-religious duty for Cypriot *ethnarchs*. The Hellenic–Orthodox essences of Greek nationalism had embraced *Megali Idea* (Great Idea), foreseeing the liberation of the territories inhabited by Hellenic–Orthodox communities from their foreign rulers and the annexation of these territories by the Greek nation state. Thus, for Greek nationalism and its version in Cyprus, the island was a Hellenic–Orthodox territory that had to be unified with the Hellenic–Orthodox "motherland" (Greece). In 1878, the island became a British protectorate; it was unilaterally annexed by Britain in 1914 and officially became a colony in 1925. In 1931, Greek Cypriots rebelled against the British for Enosis, and the rebellion was championed by the church. In 1950, over 95% of the Greek Cypriot electorate voted for Enosis in a church-sponsored plebiscite (Kıralp 2014).

In making sense of Cypriot history, a noteworthy number of scholars, including Pirinççi (2006), Kamil and Güneş (2014), Göktürk (2015), and Savrun (2017), argued that the Orthodox Church of Cyprus, and thus the *ethnarchy*, enjoyed significant social, economic, and political power in the Ottoman era and acted as the impetus of Greek nationalism and pro-Enosis sentiments on the island in the late 18th and early 19th Centuries. Additionally, it is also widely accepted by the scholarship that its traditional prestige and power enabled the *ethnarchy* to resist against British colonialism and launch the pro-Enosis nationalist revival (Frendo 1998; Panayiotopoulos 1999; Ker-Lindsay 2006; Jackson 2021). Nonetheless, some scholars have argued that, with the arrival of British rulers, the *ethnarchy* lost the privileges it enjoyed during the Ottoman era and this was among the factors leading it to commence the anti-British and pro-Enosis struggle (Hadjianastasiou 2018; Yiangou 2018).

On the contrary, based on Stavrides (2013), Louis (2013), Anagnostopoulou (2013), Varnava and Pophaides (2013), and Varnava and Michael (2013), one could conclude that Cypriot ethnarchs played different roles in different historical and political contexts.

These scholars agree with conventional wisdom that the *ethnarchy* enjoyed both important authorities and sociopolitical and economic power in the Ottoman era. They have also acknowledged that the British rule partially diminished the *ethnarchy's* sociopolitical power, and this was among the reasons motivating the church to launch resistance against colonial rulers. Nevertheless, they have pointed out that some Archbishop-*ethnarchs* tended to cooperate in harmony with the island's Ottoman and British rulers, while some others promoted Greek nationalism and Enosis. Furthermore, as noted by Varnava and Michael (2013), the *ethnarchy* launched the pro-Enosis struggle and national revival not in the Ottoman, but in the British era. Varnava and Pophaides (2013, p. 148) regarded Archbishop Kyrillos II (in office from 1909 to 1916) as the "first Greek nationalist and enosist Archbishop-*ethnarch*," emphasized that he presided an anti-British rally in 1912 by signing a resolution according to which "the national sentiments and unalterable wish of the people to unite with Greece would never be suppressed" (p. 164), and noted that in actuality, his predecessors took no noteworthy pro-Enosis actions. The main contribution of this book edited by Varnava and Michael (2013) argue that Cypriot ethnarchs played very different roles from one another and that the ethnarchy undertook different tasks in different historical periods. As a matter of fact, Makarios himself did not follow exactly the same political path throughout his career. In the 1950s, he championed the pro-Enosis struggle, and in the post-1968 era, he manifestly promoted independence. Thus, Makarios also played different roles under different political circumstances (see Kızılyürek 2005; Kıralp 2014, and Section 3 of this paper).

As regards Makarios' role as an *ethnarch*, Cassia (1982) argued that the Greek Cypriot community was not familiar with secularism and had no tradition of separating religion from politics. It was therefore not at all surprising that Makarios kept both posts as the President and the Archbishop and Greek Cypriots followed him as their religious and political leader. Ker-Lindsay (2006) noted that Makarios' traditional and religious authority as the Archbishop-*ethnarch* was quite helpful in maximizing his presidential power. Additionally, according to Varnava and Michael (2013), with him, "the *ethnarchism* of the Archbishop became gradually autonomous from that of the Church". The findings of this paper largely correspond with this postulation, as the three bishops opposed Makarios to serve as the Head of both the Church and the State, and were trying to get him to resign from his presidency. Additionally, while the three bishops were in league with the enosists, the *ethnarch* himself was following a pro-independence line. While the senior clergy were predominantly anti-communist, Makarios never banned communist parties; he tolerated their activities and cooperated with them in the Cyprus issue. Thus, the bishops were not capable of affecting his policies or political authority (see Sections 3 and 4 of this paper). One of the most prominent analyses on Makarios' *ethnarchy* is that of Anagnostopoulou (2013). Anagnostopoulou noted that Makarios' primary political goal in the 1960s was not Enosis or the abolition of the Cypriot State. Instead, according to the scholar, his main objective was *ethnarchifying* the Cypriot State. Anagnostopolou conceptualized Makarios' way of governance as the "*ethnarchic* State", and this conceptualization outstands as a significant contribution to the literature. As Anagnostopoulou put it, Makarios removed Turkish Cypriots from the main organs of the State, maximized his control over Cypriot politics, and paved the way for Greek-Orthodox dominance on the island under his leadership as the *ethnarch*. As Anagnostopoulou argued, many political actors who were opposed to Makarios were, at the same time, opposed to this *ethnarchic* rule. In 1968, when he announced that he would no longer follow the pro-Enosis line, pro-Enosis groups became manifestly anti-*ethnarchic* and tried to remove Makarios from his presidency. The findings of this paper correspond with Anagnostopoulou's analysis, as the three bishops, the Greek junta, and Grivas were dissatisfied with Makarios' *ethnarchification* of Cypriot politics and the authority he enjoyed as the Head of the State and of the Church. They asked him to resign his presidency to terminate this *ethnarchic* rule. Nevertheless, this paper points out that, Makarios' *ethnarchic* rule was an important factor capable of making direct or indirect impacts on Greek politics and Greek State–Church relations as well. For instance, the

Archbishop of Ieronymos of Greece was deposed mainly due to his pro-Makarios attitudes (see Section 4 for more details).

Makarios was born in 1913 in a mountainous village in Cyprus. His name was Michael Mouskos. His mother passed away when he was nine years old. His father delivered him to the monastery, where he was trained as a priest. At the age of 25, the church gave him the name "Makarios". He studied theology in Athens, and was planning to be a theology professor, continuing his education at Boston University. In 1948, he was appointed as the Bishop of Kitium and returned to Cyprus. In 1950, once his predecessor passed away, he was elected as the new Archbishop. At the age of 37, he assumed *ethnarchy* and became one of the most important figures in the political history of Cyprus. He regarded paving the way for Enosis and providing Greek Cypriots with national rehabilitation as his main duties. In 1952, he made a window-dressing reform concerning Church–society relations and established the "National Assembly". The "Assembly", in theory, was supposed to represent various segments within the Greek Cypriot community. In practice, however, it played no noteworthy roles apart from approving the *ethnarchy's* decisions (Mayes 1981). It is also noteworthy that, in spite of the ethnic conflict the two communities on the island suffered, Archbishop Makarios expressed, on many occasions, that he respected Turkish Cypriots' religion. According to him, religion had "never been a cause of friction", as the religious beliefs of all the communities on the island were "deeply respected" (PIO 1 1970). Furthermore, he was a close friend and a trustworthy regional partner of Egyptian leader Gamal Abdel Nasser (Kıralp 2019) and a symbol of resistance against oppression in the eyes of Palestinian Arabs (Assos 2018). Therefore, one may claim that, for Makarios, the internal aspects of the Cyprus issue constituted an ethnic conflict instead of a clash between the two religions, and his international prestige was respected by a noteworthy number of Muslims outside Cyprus.

### 3. A Communist-Friendly Modern *Ethnarch*: President Makarios

In the postcolonial era, the Orthodox Church of Cyprus remained noticeably more powerful compared to a significant number of its counterparts in the Balkans, as it was not under communist rule. Orthodox churches in the Balkans suffered a serious decline in their sociopolitical power due to the anti-religious attitudes of communism. For instance, in Romania, around 200 priests were imprisoned and six bishops forcibly retired. Patriarch Justinian desperately claimed that there had been principal similarities between Marxism and Christianity to avert further government pressure on the church. In Bulgaria, Metropolitan Stephen of Sofia forcibly retired. In Albania, the Orthodox Church was urged to convert itself into a satellite of its Russian counterpart and the clergymen opposing this tendency were deposed. In 1976, all religious places of worship were closed by the government. In Yugoslavia, properties of the Serbian Orthodox Church were confiscated and the teaching of religion in schools was prohibited (McGuckin 2010). In the postcolonial era, the Orthodox Church of Cyprus maintained its economic and sociopolitical power (Markydes 1977). On the contrary, the Greek civil war (1946–1949) led to the destruction of churches and the murder of clergymen. Furthermore, the ideological orientations of communism and the Soviet Union posed threats to religious values, and the Orthodox Church of Greece sided with anti-communists against communism (Markides 2009). Importantly, in sharp contrast with the Greek Church, Makarios sided with communists against anti-communists in his domestic and international politics.

The Orthodox Church of Cyprus was also considerably more powerful and influential in domestic politics compared to its counterparts elsewhere in the Middle East. In Jerusalem, the Christian Orthodoxy sometimes suffered due to cultural conflicts between the Orthodox people and the Orthodox Church, as the Patriarch and senior clergymen were Greeks and the Orthodox population Arabs (Roussos 2005). On the contrary, the Egyptian Orthodoxy was tiny in population and depended mostly on the government's tolerance. They occasionally enjoyed reforms enhancing the legal status of the Patriarchate and allowing the Orthodox population to build new churches on the condition that the Pa-

triarch supported the government (McCallum 2007). Unlike the Patriarch in Jerusalem, the Orthodox Church of Cyprus did not suffer due to cultural conflicts, as the overwhelming majority of worshippers and the senior clergymen were Greek Cypriots. In sharp contrast with the Egyptian Orthodoxy, the overwhelming majority of the island's population were Christian Orthodox people and the conditions for the protection of the religious freedoms of the Orthodox population and the status of the Church were spontaneously present.

As the President of the Republic, Makarios was the Chief Executive, and, during the Byzantium ceremonies held on Sundays, his voice was echoed all around the island via the State's radio. In Orthodox traditions, kissing a priest's hand is a way of showing respect to his religious duty. Greek and Greek Cypriot statesmen, including the Greek military dictator George Papadopoulos, tended to kiss Makarios' hand in their meetings. As he elicited such respect, Makarios was perceived as a religious, social, and political "father." When he survived an assassination attempt in 1970, many Greek Cypriots believed that Makarios was somehow protected by God (Markydes 1977). Peter Ramsbotham, who was the British High Commissioner in Cyprus from 1969 to 1971, expressed Makarios' charismatic authority as follows:

> On the great Byzantine church feast days in Cyprus, there was his hieratic figure, a hot, hot day, there, with all his robes on, wearing a sceptre. He was the last surviving priest-king. He was a dignified, cheerful and confident man. I remember that people used to think that they would get his assent as he was so courteous. Not a bit. Anyhow, at certain times, one could get quite close to him. He believed, I think, that he was protected by providence, and in a sense he was. I remember an extraordinary, rather miraculous escape, from assassination in 1970. One early morning I woke up (my High Commission office was not far away), to the crackle of machine-gun fire. In the mornings, he was always the archbishop and in the afternoons he was the president. (Churchill Diplomatic College 2001)

After the death of Makarios, no Greek Cypriot leader went on to holding both positions (archbishopric and presidency), and one can argue that Makarios was the last actual *ethnarch* of Cypriot Hellenism (Ker-Lindsay 2006). Nonetheless, the Orthodox Church of Cyprus is still influential in the politics of education, as well as the Cyprus issue. For instance, as a sociopolitical tradition, in the post-Makarios era, each President of the Republic sought for approval from the Church for each Minister of Education he intended to appoint. This tradition was broken for the first time by the communist party AKEL (*Anorthiko Komma tou Ergazomenou Laou*—Progressive Party of the Working People), as they refused to seek such approval when they came into power in 2008 (Sarris 2010). In 2004, the UN Secretary General Kofi Annan prepared a solution plan foreseeing the reunification of Cyprus within a federal partnership. The Greek Cypriot clergy stood against this plan, which was among the factors leading 75% of Greek Cypriot voters to vote "no" (Michael 2011, p. 178).

Makarios was the leader of the anti-colonial struggle for Enosis in the 1950s. When London refused to allow Cypriots to have the right to self-determination, George Grivas, a Greek colonel originally from Cyprus founded the EOKA (*Ethniki Organosi Kyprion Agoniston*—National Organization of Cypriot Warriors) and launched an armed struggle against the British. On 1 April 1955, the EOKA commenced its attack against British targets, and Makarios assumed the political leadership of this anti-colonial struggle. In 1956, Makarios was exiled by the British (Holland 1998). For the British, the clergy, EOKA, and AKEL were "dangerous" to the same extent, as none of them favored the continuation of the British rule on the island. The "humiliating" results of the Suez Crisis urged the British to believe that they no longer needed the whole of Cyprus (Yorgancıoğlu and Kıralp 2019). Turkey and Turkish Cypriots did not accept Enosis and they promoted the partition of Cyprus by Greece and Turkey. As the Cyprus issue damaged Greco-Turkish relations, the U.S. (United States) urged the sides to resolve their conflict based on the formula of a bicommunal independent state (Dodd 2010, pp. 20–40). Makarios, who championed the pro-Enosis struggle as the *ethnarch*, was displeased with the power-sharing design

that rendered the Greek Cypriot majority (82% of the population) politically equal to the Turkish Cypriot minority (18% of the population). Furthermore, Makarios was blackmailed by Greek Prime Minister Constantine Karamanlis, who told him that Athens would no longer support his struggle if he refused to sign the international treaties that led to the establishment of the Republic of Cyprus (Markydes 1977). After the independence, Britain attained two sovereign base areas on the island. Additionally, Britain, Turkey, and Greece became the "guarantor states" obliged to preserve the island's territorial integrity and constitutional order. The constitutional power-sharing and political equality between the two communities was ensured by veto rights (Hoffmeister 2006, pp. 4–6).

Based on the principle of bi-communality, the constitution of the Republic of Cyprus separated the two communities based on their ethno-religious identities. According to Article 2, the Greek Cypriot community comprises persons "who are of Greek origin and whose mother tongue is Greek or who share the Greek cultural traditions or who are members of the Greek-Orthodox Church". Additionally, in accordance with Article 112, it was assured that "the autocephalous Greek-Orthodox Church of Cyprus" would "continue to have the exclusive right of regulating and administering its own internal affairs and property in accordance with the Holy Canons and its Charter in force". Essentially, the Turkish Cypriot community was also provided the same religious rights, and its members were defined as persons who are of "Turkish origin and whose mother tongue is Turkish or who share the Turkish cultural traditions or who are Muslims" (Article 2). Based on Article 41, the members of the Council of Ministers, the legislative branch, municipal councils, armed and security forces, and public and municipal servants were incompatible with the offices of the President of the Republic and the Vice President of the Republic. Notably, such restrictions were not valid for the clergy, and Makarios could become the President of the Republic without resigning the archbishopric. In the political design of the Republic of Cyprus, there was no cross-ethnic voting. The President would be a Greek Cypriot and elected exclusively by Greek Cypriots. The Vice President would be a Turkish Cypriot and elected by Turkish Cypriots (European Union Agency for Fundamental Rights 2021). Under such circumstances, as he was elected by Greek-Orthodox voters, Makarios was, in practice, exclusively responsible for his own community. One might claim that this encouraged him to primarily promote Greek Cypriots' interests. Furthermore, his constitutional rights to keep both positions as the Head of State and the Head of the Orthodox Church of Cyprus simultaneously provided Makarios with authorities of a modern *ethnarch*.

In December 1959, Archbishop Makarios was elected the President of the Republic of Cyprus. He won 66% of Greek Cypriot votes. His rival, Ioannis Clerides, was supported by AKEL (Kızılyürek 2005). For Makarios and Greek Cypriots, the veto rights granted to Turkish Cypriots by the constitution would, in practice, create a "minority rule" over the majority (Kıralp 2019). According to Makarios, "nowhere else in the world the 18% of the population claimed the right to manipulate the other 82%" (Makarios 1975). Therefore, until the withdrawal of Turkish Cypriots from the executive and legislative branches of the State, Makarios and Greek Cypriots were displeased with the constitutional order (Markydes 1977). Makarios had established good relations with non-aligned countries during the pro-Enosis struggle. As an anti-colonialist leader, he was highly popular in those nations and was well aware that he could rely on them regarding the Cyprus issue (Clerides 1990). In September 1961, Makarios made Cyprus a member of the NAM and he was among its founding fathers (Markides 2001).

Alongside Turkey, the Karamanlis Government in Greece was displeased with Makarios' plans to demand constitutional amendments, as such an action was likely to damage the stability in the Eastern Mediterranean (Şahin and Topbaş 2016). On 25 November 1963, Makarios demanded a set of constitutional amendments that would terminate the political equality between the two communities and pave the way for the establishment of a majoritarian democracy that would enable the Greek Cypriot majority's will to be reflected in executive and legislative decision making. The constitutional amendments he demanded would have abolished Turkish Cypriots' veto rights and transformed the

Republic of Cyprus into a Greek Cypriot-ruled island (Salih 1978, pp. 132–43). The Turkish side promptly rejected Makarios' demands, and this was followed by a political crisis that led to inter-ethnic violence on the island (Kızılyürek 2016, pp. 319–29). Polycarpos Giorkatzis, the Minister of Interior, and Nicos Sampson, a leading member of the former EOKA, were among those leading armed attacks targeting Turkish Cypriots. Nevertheless, it was believed that Makarios had lost control over the two leaders and their paramilitary groups (Milliyet 1964).

The inter-ethnic violence urged Turkish Cypriots to form ghettos and separate themselves from the Greek Cypriot majority. Turkish Cypriot members of the executive, legislative, and judiciary left their positions. The State remained purely under Greek Cypriot control. As the guarantor states, Turkey and Greece were also involved in the conflict, which led to a severe crisis between the two NATO members. Washington promoted the deployment of a NATO peacekeeping mission force on the island. Nevertheless, Makarios rejected the U.S. proposal and invited UN (United Nations) peacekeeping forces. At this point, Makarios based his policies on two options: A "complete" Enosis (with no territorial concessions to Turkey) or a "complete" independence coupled with constitutional amendments that would abolish Turkish Cypriots' veto rights (Kıralp 2018). In April 1964, Makarios visited Athens and met Greek Prime Minister George Papandreou, who had succeeded Karamanlis. The two leaders agreed on the deployment of Greek forces on the island as a measure against potential Turkish aggression (Papandreou 2006). However, Makarios' tolerance toward the pro-Soviet party AKEL and his good relations with the USSR and the NAM made NATO anxious. Papandreou had already assured London and Washington that he would oust Makarios if the Cypriot leader tried to inhibit a NATO-sponsored solution plan for the island. In other words, the Greek forces deployed by Papandreou were at NATO's service (The National Archives: CAB/128–38–59, 'Cabinets' (CM and CC series)). Furthermore, Makarios' government instructed young Greek Cypriots to join the army, namely, the National Guard, which was predominantly officered by mainland Greeks. The officers of the National Guard were loyal to Athens, not to Nicosia (Rizas 2000). Makarios was reluctant to invite George Grivas to Cyprus, and the latter was appointed as the Commander-in-Chief of the National Guard. The U.S. government was pleased with this appointment, as the NATO states believed that Grivas was a trump card against Makarios (Ball 2010).

To reduce the Greco-Turkish tension that was jeopardizing NATO's southern flank and to dispose of Makarios, who was developing "dangerous" relations with domestic and international communists, the U.S. government prepared the Acheson Plan. According to this plan, Turkey would be given a sovereign base area on the island, Britain would keep its own bases, and the rest of Cyprus would be unified with Greece. Thus, the Acheson Plan would inherently dissolve the Republic of Cyprus. Makarios was disappointed with Athens, as Greece was negotiating the plan with Turkey without consulting the Cypriot government (Nicolet 2001). Makarios told the press that he attached no importance to such a dialogue, as it was taking place in the total absence of the Cypriot people (Fileleftheros 1964). Makarios also did not accept the establishment of a Turkish base area on the island. AKEL, the USSR, and the NAM sympathized with and supported his opposition against this NATO-sponsored solution. While Makarios became one of the most prestigious leaders of the NAM, the West became highly suspicious of him. Furthermore, Makarios enjoyed friendship with and support from Nasser and Yugoslavian communist leader Josip Broz Tito. Cuba also supported Makarios in the UN General Assembly (Kıralp 2019). Due to his opposition to NATO-sponsored plans and his good relations with the Soviet Union, Makarios was often referred to as the "Red Bishop" in Washington (Terrice 2005, p. 43). It is important to stress that communist parties were banned in Greece but not in Cyprus. Makarios did not exert such prohibitions on Cypriot communists. Additionally, trade unions enjoyed large freedoms, and this was preferable for AKEL. In the eyes of AKEL, Makarios was a symbol of resistance against NATO and right-wing Greek governments. Importantly, in the postcolonial era, the anti-Makarios opposition was composed of far right

and extreme nationalist pro-Enosis actors. Although Makarios appointed no communists in his cabinets, AKEL was satisfied with his domestic and international politics (Markydes 1977, pp. 59–65).

During the crisis of 1964, Moscow announced that it would support Makarios in the case of a NATO invasion (Hale 2000, p. 158). In April 1967, the Greek junta seized power in Athens. They tried to find a peaceful solution to the Cyprus issue and asked Süleyman Demirel, the Prime Minister of Turkey, to leave Cyprus to Greece. The junta was ready to leave a sovereign base area to Turkey if the latter accepted their Enosis demand. The Turkish government did not accept this proposal (Clerides 1990). In November 1967, the National Guard attacked a Turkish Cypriot village, which led to Turkish Cypriot causalities. Turkey asked for the immediate withdrawal of Grivas and the Greek forces from the island. As Washington convinced the junta that a Turkish military intervention was imminent, Athens fulfilled the Turks' demands (Hart 1990, p. 68).

Prior to the presidential elections of 1968, Makarios made a statement and underlined that he would no longer follow the pro-Enosis line, as he identified Enosis as a "desirable" inspiration that was not "feasible". Instead, he would launch a peace process to settle with Turkish Cypriots within a democratic and independent state structure (PIO 2 1968). After this statement, the anti-Makarios press accused Makarios of "betraying the Enosis and Hellenism" (Patri 1968). Makarios' shift toward independence, coupled with a democratic solution, was welcomed by the U.S., the USSR, the NAM, Britain, Greece, and Turkey. Not only his lack of trust in Greek governments, but also the significant increase in Greek Cypriots' welfare recorded in the post-colonial era were among the reasons urging Makarios to embrace the Republic of Cyprus. Additionally, unification with Greece under a military dictatorship would deprive Greek Cypriots of the democratic rights they enjoyed in Cyprus. More importantly, the withdrawal of Turkish Cypriots from the State organs had already transformed Cyprus into a Greek Cypriot-ruled island. These factors changed Makarios' understanding of national identity, and he characterized Cypriots as a political entity independent from Greece. Therefore, he refused to follow Athens' manipulations in Cypriot politics and asserted, on many occasions, that no foreign country had the right to make decisions on behalf of Cypriots (Kıralp 2014). Additionally, while Makarios was disliked by Greek governments, his religious and political prestige was respected by the people of Greece. Unlike the junta, Makarios had good relations with King Constantine of Greece. As a person attached to the traditions of Byzantium and Hellenic Orthodoxy, Makarios tended to express that he preferred constitutional monarchy for Greece and not republicanism. Greece's military regime was always suspicious of Makarios, as he was a close friend and potential political partner of King Constantine, who was in self-imposed exile due to his conflict with the junta. If the King was to return from exile, Makarios would become even more influential in Greek politics, which the junta was well aware of. Thus, Makarios was a leader who could play a decisive role not only in Cyprus, but also in Greece (The National Archives, FCO 9/1364, WSC 1/15, Diplomatic Report, No: 229/71, 'Archbishop Makarios III, President of Re-public of Cyprus').

As regards the changes in Makarios' understanding of nationalism, there is a considerable consensus among scholars, including Kızılyürek (2005), Loizides (2007), Peristianis (2008), and Kıralp (2014), that in 1968, Makarios shifted from Greek nationalism to Greek Cypriot nationalism. As these scholars noted, in this understanding of nationalism, a Greek Cypriot-ruled island was more preferable than Enosis, as the Greeks of Cyprus were regarded as an autonomous entity. Furthermore, Greek Cypriot nationalism attached greater importance to Greek Cypriots' interests than the Greek nation state's interests. It is also noteworthy that, as pointed out by the scholars, in the traditional understanding of Greek nationalism, Enosis was the main national inspiration for Greek Cypriots, Athens was regarded as the "national center," and Nicosia was supposed to follow it. In Greek Cypriot nationalism, however, Athens' interference in Cypriot affairs was not tolerable. Theoretically, the national identity of a society comprises a number of elements, including ethnicity, language, territory, civic ties, religion, ancestry, culture, and national duties (Smith 1991).

While the primordialist scholarship, including Shils (1957), Geertz (1963), and Van Den Berghe (1981), claims that nations are premodern entities and national identities are fixed and inherently static phenomena, modernist scholars, including Gellner (1983), Anderson (1991), and Breuilly (1993), assert that nations are products of the age of modernity and national identities are socially constructed and reconstructed. Another modernist scholar, Brass (1991), noted that political and religious elites reconstruct national identities by reinterpreting the meanings of the sociocultural determinants of national identities, including territory, language, religion, and ethnicity. As a matter of fact, Orthodoxy had always been a central value in Makarios' understanding of nationalism (Anagnostopoulou 2013). Nonetheless, he reinterpreted the "meaning" of territory, particularly in the post-1968 era, and he prioritized preserving Cyprus' territorial integrity instead of struggling for Enosis. More importantly, Greek interferences in Cypriot politics were creating danger of partition, not only because the violence conducted by pro-Athens groups was likely to provoke Turkish expansionism, but also because Athens was not at all reluctant in making territorial concessions to Turkey and achieving "double-Enosis" (Kıralp 2014). Makarios' this way of thinking is also apparent in the biographical study of Assos (2018), as the Archbishop realized that the insistence on Enosis was provoking Turkish irredentism and Turkish Cypriot secessionism. He therefore regarded independence as being a more secure and more realistic path to follow. As Loizides (2007) put it, this was converted into a political agenda within which the attachment to Greek Cypriots and Cyprus overshadowed the risky desire for Enosis.

In 1968, AKEL supported Makarios in the elections, and he gained more than 95% of the Greek Cypriot votes. The inter-communal talks produced no fruitful outcome, as the two sides could not reach a settlement on the local governance disputes. Nevertheless, inter-ethnic violence ceased until 1974. The Turkish Cypriot side accepted the set of constitutional amendments that Makarios demanded in 1963. In return, they demanded autonomy in their municipal affairs. Makarios characterized this demand as a "separatist attitude" and refused to provide Turkish Cypriots with autonomy in local governance. The Greek junta was attempting to restore its relations with Ankara. Therefore, Athens exerted noticeable pressure on Makarios to accept the Turkish Cypriots' demands (Kıralp 2017). Makarios enjoyed support from left- and right-wing political parties in the Greek Cypriot community. The anti-Makarios political actors were quite small in number and were displeased with the *ethnarch*, as he had abandoned the pro-Enosis line. In the post-1968 era, the anti-Makarios groups formed terrorist organizations (Markydes 1977, pp. 76–97). One such organization that was established in 1969, National Front (*Ethniko Metopo*), claimed that no solution apart from Enosis was acceptable for the Cyprus issue. The National Front condemned Makarios, as he tolerated AKEL. They asked Makarios to resign if he was no longer going to struggle for Enosis. For the National Front, the Greek Cypriot army and police had to receive instructions from Athens and not Nicosia (The National Archives, FCO 9/1153, WSC 1/6, Telegram, No: 20. 'National Front'). In March 1970, Makarios survived an assassination attempt. The police found signs indicating that the attackers might have been connected to the National Front, as well as Polycarpos Giorkatzis, the former Minister of Interior. One week after the assassination attempt, Giorkatzis was murdered while trying to leave Cyprus. It was widely believed that the ex-Minister was in league with an inner circle within the Greek junta and that he was murdered by his mainland Greek associates (The National Archives, FCO 9/1153, WSC 1/6, Diplomatic Report, No: 320/70, 'Murder in Cyprus: An Attempt and a Kill').

In 1971, George Papadopoulos, leader of the Greek junta, sent a letter to Makarios and asked him to accept the Turkish Cypriots' demands for local autonomy, but Makarios refused (Clerides 1990). Papadopoulos wanted to avoid conflict with Turkey and the U.S.; however, he failed to convince Makarios to moderate his stance toward the demands of Turkish Cypriots. In the same year, Grivas returned to Cyprus secretly and founded a terrorist organization named EOKA-B, aiming to oust Makarios and pave the way for a pro-Enosis leader to come into power (Uslu 2003, pp. 196–206). On 17 January 1972,

Socrates Eliades, a close associate of Grivas and a leading member of EOKA-B, met a British diplomat and asked for Britain's support against Makarios. Eliades delivered Grivas' message, which claimed that Makarios' relations with domestic and international communists were jeopardizing Western and British interests on the island. According to Grivas, the British bases on the island would be safe if Makarios was ousted and the island unified with Greece. The British gave no positive response to Grivas (The National Archives, FCO 9/1153, WSC 1/5, Telegram, No: 38. 'Grivas'). In February 1972, Makarios imported a substantial amount of machine guns from Czechoslovakia as a security measure against terrorists. The junta asserted that Makarios was jeopardizing Greek Cypriots' internal peace, as he was likely to deliver those weapons to his supporters, which would lead to a Greek Cypriot civil war. As this crisis escalated the tension between Athens and Nicosia relations, on 11 February 1972, the junta asked Makarios to resign his presidency. Makarios sent Glafcos Clerides, the Head of the House of Representatives, to the U.S. Embassy. Henry Tasca, the U.S. Ambassador to Greece, promptly met with Papadopoulos. Tasca encouraged Papadopoulos to abandon his demand for Makarios' resignation. In the eyes of Americans, forcing Makarios to resign would make him even more popular among Greek Cypriots, leading to a Greco-Turkish conflict and paving the way for further Soviet influence in the Eastern Mediterranean. Owing to the U.S. interference, Papadopoulos stepped back (The National Archives, FCO 9/1507, WSC 10/10, Telegram, No: 60. 'Cyprus'). On 16 February 1972, 3000 pro-Makarios Greek Cypriots protested foreign (actually Greek) interferences in Cypriot politics near the Presidential Palace (The National Archives, FCO 9/1507, WSC 10/10, Telegram, No: 109. 'Cyprus'). On the same day, Makarios addressed the Greek Cypriot community, emphasizing that "only Cypriots themselves ha[d] the right to make the final decision on the future of Cyprus" (Agon 1972).

### 4. The Bishops' Attempt to Oust Makarios

In July 1971, Papadopoulos expressed to Makarios that Athens would seek for a peaceful solution to the Cyprus issue by trying to settle with Turkey based on a formula that would not exclude "double-Enosis". The Greek leader also informed Makarios that he would have to resign the Cypriot presidency if he was to oppose to a Greco-Turkish agreement concerning the Cyprus issue. In September 1971, Grivas returned to Cyprus and, following the Athens–Nicosia crisis in February 1972, the junta asked Makarios to resign (Mayes 1981). As the Greek junta failed to oust Makarios, another attempt was made by Bishop Yennadios of Paphos, Bishop Anthimos of Kitium, and Bishop Kyprianos of Kyrenia in March 1972. According to Clerides, it was the Greek junta that asked the bishops to take action against the *ethnarch*. The three bishops summoned a Holy Synod and asked Makarios to resign his presidency. Furthermore, in March and April 1972, Grivas secretly met Makarios and asked him to resign. Makarios did not yield to these demands. In March 1972, Papadopoulos sent Makarios a letter and expressed that the "final say on the Cyprus problem belonged to the Greek Government as the National Centre". Makarios kept on refusing to follow Athens' manipulations in the Cyprus issue. Nevertheless, in order to restore his relations with Athens, the Cypriot President reshuffled his cabinet in June 1972 and removed three anti-junta ministers from office. This reduced the Athens–Nicosia tension, at least temporarily (Clerides 1990). After the assassination attempt in 1970, Makarios had adopted a set of security measures and, according to the three bishops, it was not acceptable for the Archbishop to lead religious ceremonies with hundreds of police officers around him. The three bishops claimed that the weapons imported by Makarios jeopardized the internal peace of Greek Cypriots. Furthermore, in the eyes of the bishops, communists were "enemies" of religion, and Makarios' tolerance toward them was unacceptable. Additionally, they claimed that the Bible and the Holy Rules forbade involvement of clergymen in politics. Based on these factors, the bishops urged Makarios to resign from his presidency (Clerides 1990). Makarios sent a letter to the bishops, asserting that the Holy Synod's decisions were invalid, as no minutes were taken and the Secretary was asked to leave the hall. Makarios noted that he had not

experienced pleasure due to his secular power and authority. Instead, he emphasized that his engagement in politics had filled him "with extreme sorrow and bitterness". According to Makarios, his involvement in politics was not contradictory to the Bible and the Holy Rules, as Greek Cypriots' struggle was not "only a struggle for their national survival, but for their religious faith as well". In his letter, Makarios also pointed out that he had received the impression that there was a "secular mastermind" behind the bishops' decision (Clerides 1990).

In the general elections of 1970, four political parties had managed to gain seats in the House of Representatives. The left-wing parties were the communist AKEL, led by Ezekias Papagiannou, and the socialist EDEK (*Eniaia Dimokratiki Enosi Kentrou*—United Democratic Central Union), led by Vasos Lyssarides. The right-wing parties were Clerides' EK (*Enosi Kentrou*—Union of the Centre) and Sampson's PP (*Proodeutiki Parataksi*—Progressive Camp). After the elections, all four parties had declared their support for Makarios' government program, as well as their loyalty to him (PIO 3 1970). The anti-Makarios party DEK (*Dimokratiko Ethniko Komma*—National Democratic Party) remained outside the House of Representatives. The popular support for anti-Makarios actors were so weak that in the presidential elections of 1973, the opposition could not even challenge Makarios by nominating an alternative candidate. Consequently, on 8 February 1973, Makarios was automatically reelected as President. Shortly after the elections, the three bishops attempted to oust Makarios. This time, the Greek government sent their ambassador to Cyprus to meet the bishops and tried to discourage them from defrocking Makarios. According to Clerides, the junta had already failed to oust Makarios, and if the bishops' attempt was to fail as well, the Greek government's prestige would be severely damaged. After the restoration of Makarios–Papadopoulos relations in mid-1972, the Greek junta did not push for the removal of Makarios from his presidency. Notably, it was not Papadopoulos but Ioannidis who used military force to oust Makarios. Furthermore, as the Greek and Greek Cypriot public were under the impression that the junta was manipulating the bishops, their failure would be perceived as the junta's failure. However, the bishops summoned another Holy Synod and defrocked the Archbishop as he refused to resign his presidency (Clerides 1990).

The Patriarchs of Jerusalem and Alexandria, as well as the Archbishop Ieronymos of Greece, condemned the three bishops' decision, claiming it had violated the canon law (NARA 1 2005). Makarios told the press that he did not recognize the Holy Synod's decision and that he would take all the necessary measures to save the Church's prestige. According to the Archbishop, the three bishops were influenced by political instigations and took a step that did not correspond with national responsibility. Makarios claimed that the decision was a reflection of the "dark medieval mentality". On the contrary, Greek diplomats told their American counterparts that Makarios was likely to defrock the bishops by calling a Supreme Synod, which would provoke Grivas and EOKA-B (NARA 2 2005). The Patriarch of Antioch also declared his support for Makarios and claimed that the bishops had violated the canon law (NARA 3 2005). Makarios also believed that the bishops' actions were invalid, as a larger synod was essential for such a decision. He told the press that a decision could only be taken by a synod composed of two representatives sent by each of the primates from Constantinople, Alexandria, Antioch, Jerusalem, and Athens. Even though the bishops' move constituted no serious threat against Makarios and he could simply ignore them, the Archbishop was well aware that the bishops' decision made many Orthodox Greek Cypriots uncomfortable. Grivas and EOKA-B were evidently on the bishops' side; when Makarios expressed that he did not recognize the Holy Synod's decision, members of EOKA-B attacked a police station (NARA 4 2005).

On 14 March 1973, Patriarch Pimen of the Russian Church declared his support for Makarios. This was followed by pro-Makarios statements by the Orthodox churches of Bulgaria, Poland, and Czechoslovakia. The pro-Makarios statements of the Orthodox churches enjoyed significant media coverage. The pro-Makarios press asked the Orthodox leaders to defrock the rebel bishops. The three bishops left for Athens and the Greek

government became anxious, as it had been planning to refrain from taking action against Makarios at that stage (NARA 5 2005). On 15 March 1973, EOKA-B attacked three police stations and also organized bombing attacks targeting the houses of seven police officers. Furthermore, the coffee shop of a Grivas supporter was destroyed by another bombing attack. There were no casualties. Makarios had intended to dismiss a number of pro-Grivas police officers, and it was believed that EOKA-B was trying to prevent Makarios by conducting violence (NARA 6 2005).

On 15 March 1973, the Council of Ministers dismissed 32 police officers. On 15 and 16 March 1973, pro-Grivas targets were hit by bombing attacks. It is significant that both sides refrained from causing causalities. On 16 March 1973, another police station was attacked. Even though Athens did not ostensibly interfere, the pro-Makarios groups strongly believed that Athens was backing the opposing camp (NARA 7 2005). On 17 March 1973, the Greek government condemned the bishops' decision to defrock Makarios. While the Athens-based pro-junta press asked the Greek government to take action and prevent intra-Hellenic bloodshed in Cyprus, a governmental building and a police officer's house became targets of bombing attacks. There were no casualties, but the attacks caused substantial material damage (NARA 8 2005).

On the following days, the government spokesman told the press that the bishops would be forgiven if they stepped back from their decision. Interestingly, Makarios utilized his spokesman, who actually represented his secular post (presidency) in dealing with a religious issue. The Ecumenical Patriarch and Greek Holy Synod refused to support the three bishops in the international synod and they remained completely isolated (NARA 9 2005). As EOKA-B was constantly attacking Makarios' supporters, the government provided Vasos Lyssarides with police protection as his life was in jeopardy. Additionally, the government decided to establish a special police force to suppress EOKA-B. The fact that this mission was led by a Cypriot officer was an indicator of Makarios' distrust toward Athens. On 30 March 1973, Makarios held a press conference and expressed that Grivas was conspiring a coup against the Cypriot government. The Archbishop also clarified that he would take action after the Orthodox Easter and that the bishops would be forgiven if they stepped back. He also noted that neither himself nor his ministers would attend the anniversary of EOKA's establishment on 1 April (NARA 10 2005). On 2 April 1973, Archbishop Ieronymos of Greece resigned due to "his health conditions." This aroused a number of questions, as the Archbishop was a pro-Makarios figure (NARA 11 2005). On 5 April 1973, an EDEK member was murdered by EOKA-B (PIO 4 1973). On 8 April 1973, a police station near a Turkish Cypriot enclave was attacked by EOKA-B. Three police officers were wounded, and one of the intruders was killed. Makarios characterized this attack as "foolishness", as it led to human loss and because it was risky to instigate combat with Turkish Cypriots (NARA 12 2005).

On 13 April 1973, the three bishops declared that Makarios was no longer a part of the clergy (NARA 13 2005). The Archbishop made it clear that he would wait until the end of the Easter holiday (29 April) to take action against the bishops. It was expected that he would call for a Supreme Synod. He was likely to demand support from the Orthodox leaders of Alexandria, Jerusalem, and Antioch. Makarios also claimed that a number of Greek officers were providing support to Grivas and asked Athens to take measures against these attitudes (NARA 14 2005). On 19 April 1973, the bishops instructed priests to omit Makarios' name from church prayers and substitute it with that of Bishop Yennadios of Paphos. There was only one case of a priest following this instruction, while the overwhelming majority of priests and Greek Cypriots were on Makarios' side (NARA 15 2005). On 25 April 1973, EOKA-B asked the priests to follow the bishops' instructions and omit Makarios' name from church prayers. The pro-Makarios press claimed that an international synod would be summoned to deal with the crisis (NARA 16 2005).

An international synod had to be composed of a minimum of 13 representatives drawn from Jerusalem, Antioch, Alexandria, Constantinople, and Greece. Makarios told the press that a synod would be summoned even if one or more churches refused to

participate. He expressed that the rebel bishops would be charged with violation of canon law. He also emphasized that unanimity was not a requirement for the synod to make its decision (NARA 17 2005). The Patriarchs of Jerusalem, Antioch, and Alexandria agreed to be represented. Makarios expected no support from the Orthodox Church of Greece, believing they would not participate. Nonetheless, he intended to invite them. He also expected that the Ecumenical Patriarchate would not join the synod, yet they would also be invited (NARA 18 2005). One month before the Supreme Synod, despite Archbishop Ieronymos's pro-Makarios attitudes, the Greek Holy Synod recognized the validity of the Cypriot bishops' defrocking of Makarios. This was completely unexpected by pro-Makarios Greek Cypriots. American diplomats reported to Washington that this move was conspired by the Greek government (NARA 19 2005). On 6 July 1973, the international synod held its meeting in Cyprus. The rebel bishops were accused of schism and conventicle. The synod comprised 16 prelates, including those representing the patriarchates of Alexandria, Antioch, and Jerusalem. The Ecumenical Patriarchate and the Church of Greece did not participate. The prelates unanimously decided that the bishops' decision was contrary to canon law and was, therefore, null and void (NARA 20 2005). On 14 July 1973, the three bishops were defrocked by the synod (NARA 21 2005).

As demonstrated by the crisis provoked by the bishops, unlike what it had done in 1972, the Greek junta avoided direct interference in 1973. Nevertheless, it was strongly believed that the junta was somehow involved in Ieronymos's resignation, as well as the Greek Synod's decision to recognize the Cypriot Synod's decision. As a matter of fact, Grivas and his terrorist organization manifestly supported the bishops against Makarios. In early 1972, the Greek junta asked Makarios to resign so the bishops' decision could religiously and politically legitimize his ouster. Nonetheless, as Athens–Nicosia relations were restored in mid-1972, in 1973, the junta refrained from supporting the bishops. While there had been a trilateral anti-Makarios league (comprising the junta, Grivas, and the Cypriot Holy Synod) in 1972, one year later, the junta abandoned its policies aimed at ousting Makarios, at least ostensibly. The *ethnarch* sacrificed three ministers by removing them from the office and managed to moderate the junta's stance against him. The bishops tried to stage a religious coup against the President. Their religious authority was abused for political purposes by the junta and by anti-Makarios terrorists in 1972, as well as by the latter in 1973. Makarios managed to save himself from the religious coup in 1973. Nevertheless, Greek Brigadier Dimitrios Ioannidis ousted Papadopoulos on 25 November 1973 (NARA 22 2005) and Makarios on 15 July 1974. Importantly, Greek Archbishop Ieronymos had resigned, yet he would practically stay in office until the election of the new Archbishop. The Ioannidis-sponsored government ignored him and asked Bishop Serafim of Ioannaina to administer oath of office. This made pro-Makarios Ieronymos feel humiliated. In the following days, he left office and was succeeded by Serafim (NARA 23 2005). It is well-known that Ieronymos was appointed by Papadopoulos, who seized power in April 1967, and his predecessor Chrysostomos II had forcibly resigned (NARA 24 2005). The pro-Makarios attitudes of Ieoronymos during the ecclesiastical coup had led to conflict and dissatisfaction among Greek senior clergymen (NARA 25 2005). One could claim that Ieronymos's sympathy toward Makarios displeased Papadopoulos and Ioannidis. Additionally, due to his pro-Makarios stance, he was isolated by the senior clergy and could not remain Archbishop.

On many occasions, the Greek junta tried to dictate policies to Makarios, who refused to receive orders from Athens. When Grivas died in January 1974, Ioannidis took EOKA-B under his direct control and encouraged the terrorists to intensify the violence against the Cypriot government. Compared to Papadopoulos, Ioannidis was far more extremist, and he did not tolerate Makarios' "disobedience". On 15 July 1974, the junta staged a coup against Makarios, and the Cypriot President left the island. Nevertheless, a few days later, he represented Cyprus as the official President of the Republic at the UN Security Council as the junta-sponsored regime lacked international recognition. In other words, he had to leave the island, but he remained Head of State. Moreover, the ouster of Makarios

paved the way for Turkish military action, which led to the division of the island into two (Kıralp 2017).

**5. Conclusions**

 This paper provided a historical background for understanding Cypriot *ethnarchy*'s political authority and examined the politics and political involvements of the Orthodox Church of Cyprus in the early postcolonial era. Its findings demonstrate that the rebel bishops manifestly cooperated with political actors, including the Greek junta and anti-Makarios extremists. The Greek junta was displeased with Makarios, as he, on many occasions, refused to follow the junta's manipulations in Cypriot politics. As Makarios was the President of the Republic and the Archbishop of the Orthodox Church, it can be postulated that the junta required not only political but also religious legitimacy while trying to oust him. When asking Makarios to resign, the junta accused him of jeopardizing Greek Cypriots' internal peace. The U.S. government discouraged Makarios' ouster simply because it would lead to a Greco-Turkish crisis and pave the way for Soviet interference. Once the U.S. interference stopped Papadopoulos, the three bishops took action and asked for Makarios' resignation. Notably, as Athens failed to oust Makarios in 1972 and Makarios removed three anti-junta ministers from office to restore his relations with Papadopoulos, the Greek junta avoided overtly supporting the bishops in their attempts on ouster of Makarios in 1973. As his prestige in the NAM and the support he enjoyed from foreign Orthodox churches demonstrate, Makarios was an internationally respected political and religious leader. While Orthodox churches in communist states such as Russia, Bulgaria, Poland, and Czechoslovakia declared their support for Makarios, the Greek Synod recognized the validity of the Cypriot bishops' decision. On the contrary, to offer assistance to the bishops, Grivas and EOKA-B intensified their violence against pro-Makarios targets. Although the opposition accused Makarios of "betraying Hellenism" due to his pro-independence policies, he enjoyed an overwhelming victory in the 1968 elections.

 The anti-communist ideological orientation was a common feature of anti-Makarios actors, including Grivas, the National Front, and the three bishops. As the findings of this research demonstrate, when asking Makarios to resign, the three bishops and the terrorist organizations made references to his tolerance toward Cypriot communism. One can hardly claim that domestic and international communists targeted Cyprus' independence or integrity. Instead, the USSR had manifestly backed Makarios during the 1964 crisis against the U.S.-sponsored Acheson Plan. Similarly, AKEL unwaveringly supported Makarios in the Cyprus issue. In the 1968 elections, Makarios increased his votes by nearly 50% owing to the support he enjoyed from AKEL. Therefore, it was quite reasonable for Makarios to tolerate Cypriot communism. Importantly, Makarios was the leader of the pro-Enosis struggle, and he experienced serious political conflicts with Greek governments in the postcolonial era. Grivas was among his associates during the anti-British struggle, but he turned against Makarios' government later. Additionally, as the Archbishop, he was defrocked by his own associates in the Church. It is also important to stress that Makarios was also capable of playing different roles in different historical thresholds. In the 1950s he was the leader of the pro-Enosis struggle; in the post-1968 era, he followed a merely pro-independence line.

 In the case studied in this paper, the Orthodox Church was manifestly and inevitably involved in politics, as Makarios was the President of the Republic and as his rivals utilized the bishops against him. It is also noteworthy that the Cypriot church might have been used as a Greek instrument in shaping Greek Cypriot politics, as the three bishops tried to force Makarios to resign immediately after the failure of the junta to oust the Cypriot leader. Additionally, the State–Church relations in Greece and Cyprus appear to have been interrelated and interdependent, as the pro-Makarios Archbishop Ieronymos could not remain in office. His pro-Makarios attitudes were disliked by Greek senior clergymen; he found himself isolated and Ioannidis treated him in quite a derogatory manner. It is

useful to stress that Makarios did not embrace the power-sharing system in which the two communities were politically equal. Nevertheless, in his eyes, the religious difference between Turkish Cypriots and Greek Cypriots was not a source of conflict, and he was popular in the Arabic realm as well.

**Author Contributions:** Conceptualization, Ş.K.; Methodology, Ş.K. and A.G.; Resources, Ş.K.; Supervision, Ş.K.; Visualization, A.G.; Writing—original draft, Ş.K.; Writing—review and editing, A.G. All authors have read and agreed to the published version of the manuscript.

**Funding:** This project received no external funding.

**Conflicts of Interest:** The authors declare no conflict of interest.

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
