# Peer review of "Ousting the Cypriot Ethnarch: President Makarios’ Struggle against the Greek Junta, Cypriot Bishops, and Terrorism"

_religions, doi:10.3390/rel12110944_

Round 1

Reviewer 1 Report

This proposed article is both timely and welcome. The author has conducted some good archival work. However, in its current form, I cannot recommend publication for three serious reasons.

Firstly, the article is not written in English that can be said to be acceptable for publication. There is no doubt, having worked with many authors who have English as a second language, that this author is thinking in one language and writing in English. This could of course be corrected by an editor working with the author.

Secondly, the author needs to be less descriptive and clearer at the start on what this article will do, how it is original and what it will add to the historiography.

The third, and more serious issue, is the failure of the author to consult texts that have grappled with the role of the Cypriot ethnarch, and with the presidency of Makarios. This does not merely mean referencing these publications, but in fact meaningfully engaging with them, stating whether the author agrees or disagrees with them, and what new they are brining to the discussion. I am mainly thinking about several chapters in the edited volume (eds.) Andrekos Varnava and Michalis N. Michael, The Archbishops of Cyprus in the Modern Age: The Changing Role of the Archbishop-Ethnarch, their Identities and Politics, Cambridge Scholars Publishing, Newcastle upon Tyne, 2013 (a book listed in over 1,100 libraries in the world, so there is no excuse about access) and several biographies of Makarios. The aforementioned edited volume has a chapter by the editors that engages very meaningfully with the changing concept and character of ethnarch across time, that the author needs to engage with (chapter Andrekos Varnava and Michalis N. Michael, “Archbishop-Ethnarchs since 1767”, (eds.) Andrekos Varnava & Michalis Michael, The Archbishop’s of Cyprus in the Modern Age: The Changing Role of the Archbishop-Ethnarch, their Identities and Politics, Cambridge Scholars Publishing, Newcastle upon Tyne, 2013, 1-16.) Then there is another chapter dealing with the subject of Makarios by Sia Anagostopoulou. As for biographies of Makarios, the author needs to consult and engage with those by Stanley Mayes and the more recent one by Demetris Assos, even if the author does not agree with these.

Author Response

Many thanks for your useful and constructive feedback. I am attaching our responses to your comments and our revisions. Best wishes. 

Reviewer 2 Report

Very well-written piece but not clear to its originality claim. I think the most interesting part involves the role of the Greek church (rest is known to the wider public). I could see a strong case for publishing this article if the focus is on this novel and lesser known aspect of (Greek) religious politics. 

Reads a little naive when it claims that 95 percent of Greek Cypriot supported Makarios (perhaps he had the majority on his side but not 95 percent)

Article lacks engagement with theory or comparative cases (e.g. breakdown of consociationalism or explanations emphasizing different understandings of nationalism).

I would also situate Cyprus across the wider spectrum of cases in Balkans and Middle East and give a bit more background on the organization of the Church of Cyprus (fairly decentralized semi-independent bishops and monasteries)

Author Response

Many thanks for your useful and constructive feedback. Attached you can see our responses to your comments and our revisions. Best wishes. 

Round 2

Reviewer 1 Report

This has significasntly improved. I think the conclusion could be stronger and look to broader conclusions and what this research might mean.

I still think some editing is required to achieve precise expression.